# Canine Adipose-Derived Mesenchymal Stromal Cells Reduce Cell Viability and Migration of Metastatic Canine Oral Melanoma Cell Lines In Vitro

**DOI:** 10.3390/vetsci11120636

**Published:** 2024-12-09

**Authors:** Fwu Shing Teng, Patricia de Faria Lainetti, Mayara Simão Franzoni, Antonio Fernando Leis Filho, Cristina de Oliveira Massoco Salles Gomes, Renée Laufer-Amorim, Rogério Martins Amorim, Carlos Eduardo Fonseca-Alves

**Affiliations:** 1Department of Veterinary Surgery and Animal Reproduction, School of Veterinary Medicine and Animal Science, São Paulo State University (UNESP), Botucatu 18618-681, Brazil; teng.fwu@unesp.br (F.S.T.); patricia.lainetti@unesp.br (P.d.F.L.); mayara.s.franzoni@unesp.br (M.S.F.); a.leis@unesp.br (A.F.L.F.); 2Department of Veterinary Pathology, School of Veterinary Medicine and Animal Science, University of São Paulo (USP), Sao Paulo 05508-270, Brazil; cmassoco@gmail.com; 3Department of Veterinary Clinic, School of Veterinary Medicine and Animal Science, São Paulo State University (UNESP), Botucatu 18618-681, Brazil; renee.laufer-amorim@unesp.br (R.L.-A.); rogerio.amorim@unesp.br (R.M.A.); 4Institute of Veterinary Oncology (IOVET), Sao Paulo 05027-020, Brazil; 5Vet Precision Laboratory, Botucatu 18610-034, Brazil

**Keywords:** dog, melanocyte, interleukins, peripheral mononuclear cells, stem cells

## Abstract

Canine oral melanoma is an aggressive cancer in dogs that may respond to new treatments designed to boost the immune system’s ability to fight the disease. In this study, we investigated whether mesenchymal stem cells derived from fat tissue could stimulate the immune system against this cancer. Mesenchymal stem cells are known to produce proteins called cytokines that help support immune functions. We tested whether combining these stem cells with another type of immune cell could inhibit cancer cell viability and migration. We also examined three key interleukins—2, 8, and 12—associated with immune responses to understand how they might reduce cancer cell growth and migration. Our findings showed that mesenchymal stem cells indeed reduced cancer cell migration and survival, although the effects varied depending on the cancer cell type, suggesting that some cells respond better than others. These results could aid in developing new, less invasive therapies that harness the immune system to fight cancer in dogs, potentially reducing the need for more aggressive treatments and improving patient outcomes.

## 1. Introduction

Canine oral melanoma (COM) is one of the most frequent and aggressive oral tumors encountered in dogs [1]. It can develop spontaneously in dogs and usually presents with aggressive behavior, local infiltration, and high metastatic rates [2,3]. Owing to its aggressive behavior, multimodal therapies are required to treat COM, including surgery [4], radiation therapy [5], electrochemotherapy [6], chemotherapy [7], and immunotherapy [8,9]. This approach has shown promising results in both human and canine melanoma, since COM is considered a “hot tumor”—characterized by a high mutation rate and a robust immune system response [10,11,12].

The tumor microenvironment utilizes multiple mechanisms to diminish the immune system’s antitumor activity. These include the activation of bone-marrow-derived suppressor cells, induction of regulatory T cells (Tregs), evasion of immune recognition through non-expression of histocompatibility molecules, and modulation of interleukin production [13]. For example, Catchpole et al. [14] previously identified the increased production of IL-10 and TGF-ß using multiplex RT-PCR in a canine metastatic lymph node, along with decreased levels of IL-2, IL-4, and IFN-γ, indicating loss of lack of immune activation.

Multipotent mesenchymal stromal cells (MSCs) exhibit a fibroblast-like morphology and possess both self-renewal and differentiation capacities, enabling them to proliferate for over 40 generations while maintaining their multipotent properties [15]. MSCs are often regarded as immune-privileged cells due to their lack of major histocompatibility complex class II (MHC-II) expression, which prevents immune cells from recognizing them as foreign and triggering an immune response [16]. Additionally, MSCs produce a range of immunomodulatory cytokines, including prostaglandins (PGE2), IL-4, IL-6, IL-10, and TGF-β [17].

Cancer has recently emerged as a promising target for cell-based therapies, with MSCs playing a potential role in modulating the tumor microenvironment through the expression of various cytokines and cellular interactions. Certain interleukins produced by MSCs can further regulate the immune response and exhibit antitumor effects by directly influencing tumor cells or activating the host immune system [18]. Specifically, MSC-derived interleukins can enhance antineoplastic immune surveillance by activating natural killer (NK) cells and cytotoxic lymphocytes, promoting an immune-mediated response against tumor cells [18]. However, the results of previous studies using MSCs to treat different tumors are divergent, demonstrating their capacity to suppress tumor progression and intensify tumor growth and metastasis [19].

MSCs can play a pivotal role and can be subdivided into pro-inflammatory (MSC1) or anti-inflammatory (MSC2) phenotypes [20]. Waterman et al. [20] identified that human MSCs, when induced to adopt the MSC1 phenotype, attenuate cancer cell growth. In contrast, MSCs induced to adopt the MSC2 phenotype promote cancer growth and dissemination, similar to conventional MSCs [21]. This highlights the critical role of the interaction between these cells and immune cells in determining the effectiveness of cell-based therapies.

COM has previously been used as a model for antitumor immunotherapy, demonstrating promising therapeutic potential by stimulating the immune system against tumor cells [22]. Cellular therapy using MSCs for neoplasm treatment has been studied, showing promising results in various types of neoplasms, including human melanoma [23,24]. The inhibitory potential of MSCs has previously been demonstrated both in vitro [25,26,27,28] and in vivo [25,29]. However, to the best of our knowledge, no previous studies have yet demonstrated the inhibitory potential of canine MSCs on canine melanoma cell lines.

This study evaluated the potential inhibitory effects of mesenchymal stromal cells (MSCs) on melanoma cancer cell lines (MCCLs), given that leukocytes are known to play a significant role in the tumor microenvironment of both human [30,31,32,33] and canine melanomas [34,35,36]. Additionally, the study examined whether MSCs modulate the effects of peripheral blood mononuclear cells (PBMCs). The investigation focused on the indirect effects of adipose-derived MSCs (Ad-MSCs), both alone and in combination with canine PBMCs, on the viability and migratory behavior of canine oral melanomas.

## 2. Materials and Methods

All procedures were approved by the Ethics Committee on Animal Use (CEUA) of the School of Veterinary Medicine and Animal Science at São Paulo State University (CEUA/UNESP #0153/2020).

### 2.1. Cancer Cell Lines and Culture

In this study, we used two MCCLs developed from metastatic COMs. Lines produced from metastatic melanoma were specifically used because previous studies have indicated that metastatic lymph nodes lack immune activation [16]. The first cell line (UNESP-MEL3) was developed and characterized by our research group using a sample obtained from a 10-year-old Teckel diagnosed with metastatic oral melanoma. This cell line was established from affected submandibular lymph nodes. The second cell line, MeLn, was acquired from the Laboratory of Comparative Immuno-oncology at the Department of Pathology, School of Veterinary Medicine and Animal Science, University of São Paulo (USP), and was similarly derived from submandibular lymph nodes affected by metastatic canine oral melanoma (COM) [37]. Both cell lines exhibited tumorigenicity when grown in vivo in xerographic nude mice (BALB/c). Further details on each cell line can be found in Appendix A.

Melanoma cells were cultured in Dulbecco’s Modified Eagle Medium/Nutrient Mixture F12 (DMEN F12) (Lonza Inc., Allendale, NJ, USA) supplemented with 10% fetal bovine serum (FBS) (LGC Biotecnologia, Cotia, São Paulo, Brazil), 1% antibiotic, and antimycotic (Gibco^®^, ThermoFisher Scientific, Waltham, MA, USA) and incubated in a humidified environment with 5% CO_2_ at 37 °C for cell expansion. Before the experiments, the cell lines were authenticated by Short Tandem Repeat (STR) DNA genotyping. The analysis was performed at the Comparative and Translational Oncology Laboratory of São Paulo State University (Pirassununga, Brazil). We further tested our cell line for mycoplasma contamination using PCR, which proved that both cell lines were negative.

### 2.2. Ad-MSC Culture and PBMC Isolation

Commercial and previously characterized Ad-MSCs from OMICS Biotecnologia Animal were used in experiments [35]. Adipose tissue samples were collected from healthy females undergoing elective ovariohysterectomy.

Ad-MSCs have a mesenchymal origin due to the expression of the surface antigens CD29+, CD44+, CD45-, CD14-, CD34-, and HLA-DR, as determined by flow cytometry, and confirmed tri-lineage potential (osteogenic, adipogenic, and chondrogenic differentiation in vitro) [38]. After acquisition, Ad-MSCs were cultured following the protocol used for melanoma cells.

For PBMC isolation, a pooled sample of canine peripheral blood was collected in an EDTA tube and processed using Histopaque 1077 (Sigma Aldrich, St. Louis, MO, USA) following the manufacturer’s protocol. To verify successful isolation, a smear of the isolated cells was prepared and stained with Giemsa, and the morphology of the mononuclear cells was confirmed by microscopic examination.

### 2.3. Co-Culture

To define the individual and combined effects of Ad-MSCs and PBMCs on melanoma cells within a co-culture model, five groups were established, including a control group, utilizing two distinct oral melanoma cell lines (Figure 1). The first two groups were used to define the individual effects of Ad-MSCs and PBMCs on MCCLs.

The last two groups were designed to evaluate whether Ad-MSCs could modulate the effects on melanoma cells, given that these cells have previously demonstrated significant immunomodulatory properties [16,39]. Gornostaeva et al. [40] have shown that direct contact between PBMCs and Ad-MSCs enhances PBMC viability. Consequently, the third group assessed the combined effects when these cells interacted directly, while the fourth group analyzed whether the interaction between Ad-MSCs and MCCLs could influence the interaction and behavior of PBMCs by altering the conditioned medium.

For the gene expression experiment, controls consisted of Ad-MSCs and PBMCs cultured either together or separately without the presence of melanoma cell lines. In the co-culture experiments, cells were physically separated using transwell inserts (0.4 µm, ThinCert^®^, Greiner Bio-One, Kremsmünster, Austria), which permitted medium exchange without direct cell–cell contact for viability assays. For the migration assays, inserts with a pore size of 8 µm (ThinCert^®^, Greiner Bio-One) were utilized.

The ratio of MCCL: Ad-MSC: PBMC was 3:3:1. For this ratio, we decided to use the smallest proportion between MCCL:Ad-MSC previously demonstrated to exhibit an inhibitory effect in melanoma cells, the human melanoma cell line [41], and PBMC. To better simulate the proportions found within the tumor microenvironment, we decided to use cells three times lower than tumor cells. Planning was performed for each experiment to ensure that the cells remained under the same conditions. All cultures were maintained in a humidified incubator at 37 °C with 5% CO_2_ for 24 h.

### 2.4. Cell Viability Assay

Melanoma cells were seeded into 24-well plates and incubated for 24 h using 600 µL of cell suspension at a concentration of 0.83 × 10^5^ cells/mL per well. To investigate the effect of Ad-MSCs and PBMCs on melanoma cell viability, a 0.4 µm transwell insert (0.4 µm PET, Millipore, Darmstadt, Germany) was placed in the wells. Depending on the experimental group, Ad-MSCs (50 × 10^3^ cells per well) and PBMCs (16,600 cells per well) were subsequently added. After 24 h of co-culture, cell viability was assessed using the MTT assay. This method relies on the ability of metabolically active cells to convert the yellow substrate 3-(4,5-dimethylthiazol-2-yl)-2,5-diphenyltetrazolium bromide (MTT) into formazan, a purple-colored product, which was then quantified by measuring the colorimetric change [42].

Tests were performed in experimental duplicate, and cells were divided into four groups with a control for each MCCL (Figure 1—Cell viability assay). After 18 h, for group 4, the conditioned medium was saved, and the membrane insert was replaced with a new one without cells. PBMCs were then added to the conditioned medium and all groups were cultured for an additional 6 h.

In the final step, the insert for all groups was removed, and MTT was added to each well to a final concentration of 0.5 mg/mL and incubated for 4 h at 37 °C. The supernatant was discarded, and 200 µL of dimethyl sulfoxide (DMSO) (D8418, Sigma Aldrich Co. LLC., Saint Louis, MO, USA) was added to dilute the MTT salt.

Absorbance at 570 nm was measured using a microplate spectrophotometer (Sunrise, Tecan Trading AG, Switzerland). Growth inhibition rates were calculated according to the optical density (OD) values using the following formula: inhibitory ratio = (OD570 value of control group OD570 value)/control group OD572 value).

### 2.5. In Vitro Analysis of Cell Migration

A transwell assay was used to evaluate cell migration capacity (ThinCertTM, Greiner Bio-One, Kremsmünster, Austria), in accordance with the manufacturer’s instructions. Briefly, all cells were cultured under the conditions described above. A sample (100 µL) of the solution containing each cell culture was placed on 8 µm porous membrane inserts (Greiner Bio-One) at a concentration of 0.5 × 10^6^ cells/mL; the cells in the upper compartment were cultured without fetal bovine serum (FBS) (LGC Biotecnologia, Cotia, São Paulo, Brazil). Each insert was placed in a well of a 24-well plate containing medium with 10% FBS, Ad-MSCs, PMBC, or both, according to the group (Figure 1—Migration assay).

Each experiment was performed in triplicate. After 18 h, for Group 4, mechanical removal of Ad-MSCs from the well was performed, maintaining the conditioned medium and adding PBMCs. All groups were cultured for an additional 6 h. Subsequently, the conditioned medium from all groups was collected and stored in Eppendorf tubes at −20 °C, and the inserts were removed and immersed in methanol to fix the cells. Next, we stained the Transwell insert with Giemsa (Sigma Aldrich, Merck, Darmstadt, Germany). Evaluation was performed using an inverted optical microscope by counting the number of migrated cells in four high-power fields (40×) photographed for each replicate. The average number of migrated cells per replicate was calculated and used for statistical analysis.

### 2.6. Real-Time PCR

We chose three interleukins for further investigation by real-time PCR. First, IL-2 was selected as an interleukin with immunomodulatory and antitumor capabilities that is used therapeutically [43,44,45]. Further, we assessed IL-8 [28,46,47,48], which has immunosuppressive activity and is associated with poor prognosis, tumor progression, and migratory capacity, and IL-12 [49,50,51], which has immunomodulatory properties and is known to reduce tumor growth and metastatic capacity.

For the gene expression experiment, MCCL were collected, and mRNA was extracted using the RecoverAll™ Total Nucleic Acid Kit (Ambion, Life Technologies, MA, USA), after which the amount of mRNA was quantified using a spectrophotometer (NanoDrop ND-1000, Thermo Fisher, Waltham, MA, USA).

After mRNA extraction, cDNA synthesis and qPCR were performed, as previously described, using previously published endogenous genes (HPRT and GAPDH) for quantifications [52]. The interleukin sequences of the primers (*IL-2*, *IL-8*, and *IL-12*) used are described in Table 1, and the endogenous genes have been previously published [52]. The qPCRs for *IL-2*, *IL-8*, *IL-12*, GAPDH, and HPRT genes were conducted in a total volume of 10 μL containing the Power SYBR Green PCR Master Mix (Applied Biosystems; Foster City, CA, USA), 1 μL of cDNA (1:10), and 0.3 μM of each primer.

The reactions were performed in duplicate in 96-well plates using a QuantStudio 12 K Flex Thermal Cycler (Applied Biosystems, Foster City, CA, USA). The amplification reaction conditions for all primers were 40 cycles of 15 s at 94 °C and 1 min at 60 °C. The specificity of the PCR products was determined using a dissociation curve for all experiments. Relative gene expression was quantified using the 2^−ΔΔCT^ method.

### 2.7. Data Representation and Statistical Analysis

All experimental data were analyzed using the GraphPad Prism software (version 8; GraphPad Software Inc., San Diego, CA, USA). Data are presented as the mean ± standard error of the mean (SEM). The statistical significance of mean differences among multiple sample groups was assessed using Tukey’s test following a one-way ANOVA. To evaluate the relationships among all variables within each group, a multivariate correlation matrix was applied using the mean value of each group across different cell types. Correlation coefficients (r) were categorized as follows: weak (0–0.29), low (0.3–0.49), moderate (0.5–0.69), strong (0.7–0.89), and very strong (0.9–1.0), considering both positive and negative correlations. Statistical significance was defined as *p* < 0.05. All *p*-values are in Appendix A.

The heatmap and clustering were generated using the Morpheus web tool (https://software.broadinstitute.org/morpheus/ accessed on 10 November 2024). Data were normalized using Z-scores, and clusters were constructed through hierarchical clustering or k-means, with “one minus Pearson correlation” as the chosen distance metric. An interleukin interaction analysis was performed using the STRING web tool (https://string-db.org/ accessed on 10 November 2024) [53].

## 3. Results

### 3.1. Cell Viability and Migration Assays

In the cell viability assay, different responses were observed between the two metastatic canine oral melanoma cell lines. MeLn showed decreased cell viability in all groups compared to that in the control group (Figure 2A). The migration assay revealed a reduction in the migration of MeLn cells across all experimental groups compared to the control group. Notably, the group treated with Ad-MSCs (G2) exhibited a significant decrease in migration relative to the group treated with PBMCs alone (G1) and those treated with both Ad-MSCs and PBMCs (G3) (Figure 2B).

In contrast, UNESP-MEL3 cells did not exhibit any statistically significant difference compared to the control group (Figure 2C), with neither Ad-MSCs nor PBMCs altering the viability or migration capacity of this cell line (Figure 2C,D).

### 3.2. Interleukins 2, 8, and 12 Gene Expression

The gene expression analysis revealed notable differences in the expression of *IL-2*, *IL-8*, and *IL-12* across both cell types.

In the MeLn cell line, *IL-2* expression was significantly elevated in groups G1 and G4 compared to the control, indicating an upregulation of this interleukin under these conditions. *IL-8* levels increased in all treated groups relative to the control; in particular, the G3 group exhibited a particularly marked upregulation of IL-8. For *IL-12*, expression levels were elevated in groups G1, G3, and G4, with G3 and G4 showing substantially higher increases (Figure 3A).

In the UNESP-MEL3 cell line, *IL-2* expression showed a statistically significant difference only between groups G1 and G2, without notable changes in other groups compared to the control. For *IL-8*, increased expression was observed only in groups G2 and G3 relative to the control. *IL-12* expression, however, was significantly elevated across all groups compared to the control, with G1 showing higher expression levels than G2 and G4, while G3 displayed the highest expression among all groups (Figure 3B).

### 3.3. Multivariate Data Analysis

First, we identified the relationships between the proteins associated with the detected genes (*IL-2*, *IL-8*, and *IL-12*) and analyzed the related pathways (KEGG Pathways and Biological Processes) using the STRING tool (Figure 4A). These interleukins are primarily involved in the cytokine–cytokine receptor interaction and cancer-related pathways (Figure 4B), serving as key mediators in signal transduction, immune response, and the regulation of activated T cells (Figure 4C).

Hierarchical and k-means clustering analyses were performed across groups for both cell lines, assessing all parameters in this study (cell migration, viability, and interleukin gene expression levels). Results indicate that interleukins clustered closely together, while migration and viability formed another group (Figure 4D).

Regarding the evaluated groups, G2 from both cell lines demonstrated similar behavior grouping with G3 of UNESP-MEL3. G1, G4, and G3 of MeLn exhibited similar patterns, characterized by increased *IL-8* and *IL-12* expression and decreased tumor cell viability and migration. Control groups for both cell lines, as well as G1 and G4 of UNESP-MEL3, were closely related, maintaining stable viability and migratory capacity, with lower interleukin expression.

In a four-cluster analysis (k = 4), the MeLn control group emerged as a distinct cluster, marked by the lowest *IL-12* expression among all groups.

Subsequent analyses involved individual k-means clustering for each cell line to assess which groups exhibited similar behaviors, followed by a multivariate Pearson correlation matrix to identify parameters with strong and statistically significant correlations.

In the MeLn cell line, treated groups are clustered closely together and distinctly from the control group when grouped with k = 2, with G3 emerging as the most distinct within the treated groups in the k = 3 clustering due to its elevated *IL-8* expression (Figure 5A). In the correlation analysis, although an overall correlation pattern was observed (Figure 5B), only the relationship between cell viability and migration was statistically significant, displaying a strong positive correlation (Figure 5C).

The UNESP-MEL3 cell line showed a contrasting pattern, with the control, G1, and G4 groups clustering closely, while G2 and G3 clustered together at k = 2 but exhibited differences in *IL-12* expression (Figure 5D). Analysis of parameter correlations revealed an inverse pattern compared between cell lines; for instance, the correlations between *IL-2* and viability or migration were positive in MeLn and negative in UNESP-MEL3, though neither reached statistical significance (Figure 5E). Nonetheless, a strong positive correlation was consistently observed between migration and viability in both cell lines (Figure 5F). Additionally, a strong negative correlation was found between *IL-8* expression and migration in the UNESP-MEL3 cell line (Figure 5G).

## 4. Discussion

In recent years, advances have been made in the therapeutic protocols applied in veterinary oncology. As in human medicine, the importance of individualized and specific treatments for each patient has begun to be emphasized [54]. This is because the same type of neoplasia can exhibit different expressions of “hallmark” genes [55], rendering each tumor unique, necessitating treatment for these individual characteristics. Because the use of Ad-MSCs in canine cancer patients is ambiguous, in the present study, we cultured melanoma cancer cells with Ad-MSCs to determine whether this association could provide an indirect antitumor response (mediated by the secretome). Our focus was to investigate whether melanoma cancer cells could stimulate both Ad-MSCs and/or the secretome of PBMCs with cytokines that influence cell viability and migration.

We proposed viability and migration assays using two COM cell lines with metastatic in vivo characteristics—one low-passage cell line (UNESP-MEL3) and one high-passage cell line (MeLn)—in a co-culture model based on a physical barrier (preventing cell-to-cell interaction) over a single treatment period (24 h). Therefore, we only evaluated the indirect effects of these cells in COM cell lines at a single time point (24 h), which can be considered a limitation of our study.

Different results were obtained for the two COM cell lines. Possible explanations for this are that MCCL possesses distinct intrinsic characteristics that may or may not contribute to the effectiveness of immunotherapy and cellular therapy with Ad-MSCs [13,56] or that the passage status may have affected the results, as prior research has reported changes in gene and protein expression levels in cells at a high passage [57], meaning that low-passage cells are more closely related to cancer cells in vivo [28].

Transcriptomic studies are essential to further delineate the molecular landscape of canine oral melanoma (COM). For instance, Proteau et al. (2022) [58] identified two distinct molecular subgroups within COM, each with potential implications for targeted therapies. One subgroup exhibited differentially expressed genes associated with immune-related pathways, suggesting that certain COMs may have increased susceptibility to immune responses, characterizing them as “hot tumors”. It has also been established that therapeutic success may depend on the intrinsic factors of the neoplastic cells, as is the case with immunotherapy [13,56].

Similarly, a study on human MSCs demonstrated that MSCs with distinct phenotypes can play different roles in tumor progression and are directly involved in immunomodulatory activity [20,21]. In our study, we used a single MSC cell line that exhibited results more closely aligned with the MSC1 phenotype, showing the ability to reduce tumor growth and contribute to immune activity. However, the association of this phenotypic pattern observed in human MSCs has not yet been established in veterinary medicine. Further studies are needed to characterize this relationship and determine whether a similar response occurs in veterinary species.

While UNESP-MEL3 did not show significant differences from the control in any group, the MeLn cell line demonstrated higher susceptibility across all experimental groups. Although no synergistic effects were observed in groups treated with Ad-MSCs and PBMCs (G3 and G4), which resulted in the clustering of all treated groups in proximity. Notably, G1 and G3—both groups exposed to PBMCs every 24 h—and G2 and G4—both initially treated only with Ad-MSCs—exhibited similar outcomes in this assay. This pattern suggests that the 24-hour exposure period may not be sufficient to elicit synergistic or antagonistic effects, and alternative time intervals should be explored.

Decreased cell viability and migratory capacity were observed in all treated groups of MeLn cells, with cell migration notably attenuated following treatment with Ad-MSCs for 24 h (G2). In the UNESP-MEL3 cell line, a reduction in migratory capacity was also observed in G2, although this decrease did not reach statistical significance in UNESP-MEL3 cells (*p* = 0.06). Results from the transwell migration assay further support that Ad-MSCs may inhibit cell migration in both MCCLs in vitro, as a significant decrease was noted in G2 for each MCCL. These findings suggest that Ad-MSCs could reduce cell migration independently of intrinsic differences between the MCCLs.

In the UNESP-MEL3 cell line, a migration decrease was noted in G2, while in MeLn cells, G2 displayed a more pronounced inhibitory effect compared to other groups. This observation suggests that interactions between Ad-MSCs and immune system cells in G3 may exert an antagonistic effect, potentially mediated by the increased IL-8 expression observed in this group. Furthermore, the exposure duration to Ad-MSCs may influence the response, as evidenced by the lack of migration inhibition in UNESP-MEL3 cells in G4, aligning with findings reported by Miloradovic et al. [26].

Treatment with isolated Ad-MSCs (G1) and PBMCs (G2) demonstrated promising effects on tumor viability and migration. This finding was unexpected, as we initially hypothesized that the G3 group—where Ad-MSCs and PBMCs interacted with cancer cell lines to simulate the tumor microenvironment—would best reflect in vivo conditions.

Interestingly, in both clustering analyses, Group 3 exhibited a unique profile compared to the other groups. These findings suggest that when Ad-MSCs and PBMCs are combined, more complex interactions occur, notably affecting the expression profiles of interleukins 8 and 12, potentially due to critical changes within the tumor secretome.

For this reason, it is important to emphasize that these assays were conducted using physical barriers (inserts) between Ad-MSCs/PBMCs and MCCLs that prevented physical cell-to-cell interactions. Therefore, these results demonstrate the inhibitory potential, probably due to the alteration of the conditioned medium pattern, including the cytokine patterns.

Therefore, it should be emphasized that the three interleukins analyzed do not explain all the results of the viability and migration assays, as no correlation was observed between these interleukins and cell viability or migration. However, in the UNESP MEL3 cell line, a correlation with tumor migration was detected. Further analyses of other interleukins will be necessary to better understand their effects. For example, other pathways have been associated with the reduction in growth or in vitro migration of human and canine melanomas by using different MSCs, such as IFN-β-overexpressing [59], inhibiting PI3K/AKT signaling pathway [60], and inhibiting expressions of NF-jB signaling [41], but were not investigated in this study. The three interleukins chosen for analysis had different effects on tumor and immune system cells.

Moreover, the lack of significant results in groups treated with PBMCs across both assays may be attributed to the use of PBMCs from healthy donors. Previous studies have indicated that only PBMCs sourced from patients in melanoma remission or progression effectively reduced in vitro migration of human MCCLs [30]. Nevertheless, our findings showed some effects in PBMC-treated groups in the MeLn cell line, suggesting that cell-line-specific responses may also influence these outcomes.

It is also important to consider that the MSC−PBMC interaction occurs bidirectionally. While MSCs can exhibit immunosuppressive or pro-inflammatory capacities, the different profiles of immune cells present (e.g., immunosuppressive cells such as regulatory T cells or antitumor cells like CD8+ T cells or NK cells) play distinct roles in MSC behavior [61]. Therefore, studies focusing on the immune status of these patients and the influence of tumor-infiltrating immune cells on the role of MSCs in tumor progression are essential.

IL-2 has previously been used as an immunotherapeutic agent for melanoma in the United States [62]. At low doses, IL-2 stimulates Tregs and inhibits antitumoral immune responses. However, at high doses, it stimulates the proliferation of NK, T, and B cells, allowing for a more efficient antitumoral response [43]. Additionally, IL-2 induces the proliferation and differentiation of T cells into effector T cells, increases the cytotoxicity of natural killer cells, and induces B cell proliferation [43].

IL-2 is currently approved for the treatment of human patients with melanoma and renal cell carcinoma [63] and is considered an important interleukin associated with immune system activation [64]. Previous clinical studies have similarly demonstrated that intravenous infusion of IL-2 increases immunity, meaning that it could be considered an important immune therapy for dogs [65,66]. In addition, intratumoral IL-2 injection has been shown to be safe and to exert clear antitumor effects in dogs with mast cell tumors [67].

Tellado et al. (2023) [68] previously used an IL-2 plasmid vector for gene electrotransfer associated with electrochemotherapy, administering IL-2 intramuscularly in dogs with stage III and IV oral melanomas. The authors concluded that electrochemotherapy plus IL-2 gene electrotransfer and IL-2 systemic administration improved patient outcomes by decreasing in vivo tumor growth. Therefore, further investigation is needed to assess IL-2 levels in canine oral melanoma samples and to determine the association between IL-2 administration and survival. IL-2 has previously been administered to dogs with acceptable toxicity [65,66,68]. However, prospective clinical trials are required to clarify whether IL-2 could represent a therapeutic option for COM.

In the present study, we observed an increase in IL-2 expression in the G1 and G4 groups of the MeLn cell line compared to the control, indicating that PBMCs alone, even after prior treatment with Ad-MSCs as applied in G4, could stimulate this interleukin’s overexpression. Furthermore, this finding suggests that the presence of Ad-MSCs alongside PBMCs (G3) may have interfered with IL-2 expression. In the UNESP-MEL3 cell line, Ad-MSCs (G2) showed a decrease in IL-2 expression compared to the group treated only with PBMCs (G1).

This result suggests that Ad-MSCs may negatively affect IL-2 expression, potentially impacting immune response regulation and intratumoral lymphocyte activation. Additionally, no correlation was found between IL-2 and cell migration or viability in either cell line, and IL-2 showed the least variation across treatment groups.

IL-8 is a protumorigenic cytokine [62,63,64], which has also been found to be associated with cancer promotion in other canine tumor models [69,70]. This interleukin can promote immunosuppressive effects in the tumor microenvironment [47,71]. However, it can also activate neutrophils and lymphocytes. Further, an increase in IL-8 is associated with tumor progression and migratory capacity [72], activating invasion and angiogenesis [73], and is considered a negative prognostic factor for some tumors [74,75].

IL-8 expression was elevated in all MeLn-treated groups, especially in G3, while in UNESP-MEL3, the increase was observed only in treatments with Ad-MSCs for 24 h (G2 and G3). Moreover, Ad-MSCs can increase the production of different interleukins such as IL-8 [28].

This may be involved in the immune resistance mechanisms of neoplasms [71]. As such, MeLn, which exhibits immunogenicity and sensitivity (decreased in viability and migration assay) to the presence of immune cells (G1, G2, and G3), may have developed some resistance to these cells. In contrast, UNESP-MEL3, which was not sensitive to the same groups, was resistant to this antitumor activity.

Although we detected a strong negative correlation between IL-8 and cell migration in UNESP MEL3, this result should be interpreted cautiously, as biologically, increased expression of this interleukin is often associated with poor prognosis [46,73,76], associated with tumor progression and migratory capacity [72]. Preisner et al. [28] previously reported that co-culturing human melanoma cell lines with MSCs led to elevated IL-8 expression, alongside enhanced migration and invasion capabilities due to angiogenic factors and metalloproteinase induction. Nevertheless, G2 showed reduced tumor migration despite the increase in IL-8 levels.

IL-12 is an endogenous cytokine that plays a crucial role in activating antitumor immune responses. As such, it is instrumental in promoting defense mechanisms against tumors, including the formation of long-lasting antitumor immune memories [77,78].

Interleukin-12 exhibited the most diverse changes among the groups. In the MeLn cell line, IL-12 expression increased only in groups treated with PBMCs (G1, G3, and G4), whereas UNESP-MEL3 showed elevated IL-12 expression across all groups.

Notably, a positive synergy between Ad-MSC and PBMC presence (G3) was observed in both cell lines, suggesting a potential effect worth exploring further, given that IL-12 is responsible for activating Th1 lymphocytes, which is a crucial process in the antitumoral response and induces the production of IFN-α by lymphocytes [50,79]. As a consequence, it leads to a decrease in the tumor’s growth and metastatic capacity [50]. Additionally, Eisenring et al. [80] previously demonstrated that IL-12 exerts a paracrine effect on B16 melanoma tumor cells, directly inhibiting their growth.

Furthermore, the concurrent increase in IL-12 expression alongside IL-8 overexpression may suggest a cellular mechanism to regulate IL-8’s proinflammatory effects, or conversely, as previous studies have shown that repeated IL-12 administration in human patients results in a sustained increase in IL-8 levels [76].

Interestingly, IL-12 has been well studied as an immunotherapy in the field of human oncology [77,78]; however, few studies have investigated the role of IL-12 in animals [81,82,83,84]. Therefore, IL-12-based gene therapy may represent a promising treatment option for dogs with COM.

Furthermore, it should be noted that the efficacy of antitumoral immune action often depends on cell-to-cell interactions. Ad-MSCs can decrease the viability and migratory capacity of COM tumor cells by altering interleukin patterns or other pathways.

The factors influencing the increase or decrease in these interleukins and their interactions with immune activity should be evaluated to understand the impact of this alteration on the direct activity of the immune system and to assess the possible use—or not—of Ad-MSCs as adjuvants in other therapy modalities, immunotherapy, and chemotherapy.

The factors driving changes in gene expression patterns related to cancer-associated and antitumor immune response pathways when using MSCs should be thoroughly analyzed to understand the impact of this therapy on neoplastic cells and their microenvironment, particularly regarding immune system interactions. Understanding these dynamics will also aid in evaluating the potential of Ad-MSCs as adjuvants in other therapeutic approaches, including immunotherapy and chemotherapy.

The therapeutic use of Ad-MSCs is promising and has shown interesting in vitro results in tumor progression and migration [24], indicating that it is a potentially interesting therapy for different neoplasms [19]. However, it is essential to encourage new studies that observe cell–cell interactions, particularly concerning the immune system, as these cells have high immunomodulatory potential [17], which can trigger different effects when introduced into the tumor microenvironment. It is also important to evaluate the use of these cells as adjuvants to chemotherapy because they can alter different pathways of tumor progression, allowing changes in protocols, such as dosage reduction or treatment duration.

## 5. Conclusions

Modulation of the tumor microenvironment may have therapeutic benefits for COM, as it indirectly reduces in vitro cell viability and inhibits in vitro cell migration of metastatic canine oral melanoma cells. However, it is necessary to understand the relationship between these cells and the rest of the tumor microenvironment, particularly the infiltrating immune cells. It is also essential to identify intrinsic factors unique to each melanoma that may impact the efficacy of this approach. Further research is needed to explore the immunomodulatory roles of MSCs and their effects on the COM microenvironment.

## Figures and Tables

**Figure 1 vetsci-11-00636-f001:**
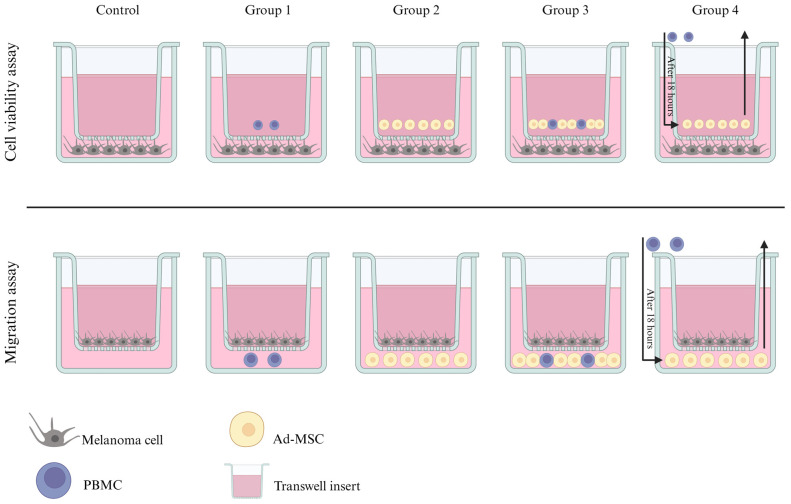
Schematic representation of the experimental groups used in the viability and migration assays. Group 1 (G1) comprised MCCLs treated every 24 h with PBMCs; Group 2 (G2) comprised MCCLs treated every 24 h with Ad-MSCs; Group 3 (G3) comprised MCCLs treated every 24 h with both PBMCs and Ad-MSCs in the same insert; Group 4 (G4) comprised MCCLs treated every 18 h with Ad-MSCs, replaced by PBMCs after 6 h. The insert in this experiment did not allow direct cell communication, and the cells were physically separated using transwell inserts (0.4 µm, ThinCert^®^, Greiner Bio-One, Kremsmünster, Austria).

**Figure 2 vetsci-11-00636-f002:**
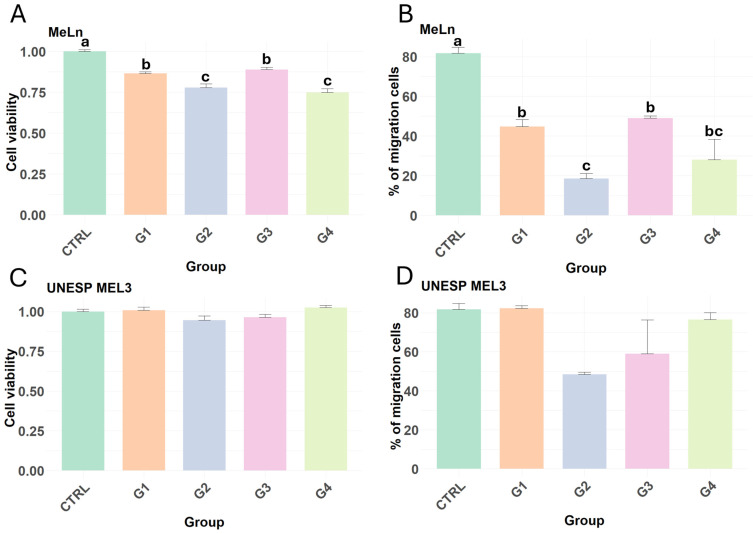
Results of cell viability and migration assays of MeLn and UNESP-ML3 cell lines. Different lowercase letters indicate statistical differences. (**A**) In the MeLn cell line, all of the treated groups showed decreased viability compared with the control group. G1 vs. G2 (*p*-value < 0.05); CTRL vs. G1; CTRL vs. G3; G1 vs. G3 (*p*-value < 0.008); CTRL vs. G2; CTRL vs. G4 (*p*-value < 0.0001). (**B**) All groups showed reduced migration in comparison with controls for the MeLn cell line. CTRL vs. G1; CTRL vs. G2; CTRL vs. G4 and G3 vs. G4 (*p*-value < 0.0001); CTRL vs. G3; G1 vs. G4 and G2 vs. G3 (*p*-value ≤ 0.0005); G1 vs. G2 (*p*-value = 0.0066). (**C**,**D**) The UNESL-ML3 cell line showed no reductions in viability and migration capacity in all groups.

**Figure 3 vetsci-11-00636-f003:**
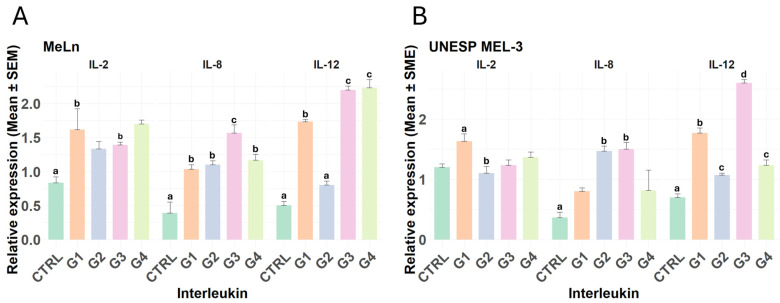
Relative expression of interleukins 2, 8, and 12. Different lowercase letters indicate statistical differences. (**A**) In the MeLn cell line, only the groups treated with PBMCs at any point (G1 and G4) showed increased IL-2 expression. CTRL vs. G1; CTRL vs. G4 (*p*-value < 0.05). Conversely, all groups exhibited increased IL-8 and IL-12 expressions compared to the control (all *p*-values are in Appendix A), with a particularly pronounced increase in IL-8 in G3 and in IL-12 in both G3 and G4. (**B**) In the UNESP-MEL3 cell line, IL-2 expression remained unchanged across all treatment groups when compared to the control, although G1 showed higher expression compared to G3. IL-8 expression was elevated only in groups treated with Ad-MSCs every 24 h (G2 and G3), while IL-12 expression increased across all groups, with the highest levels observed in groups treated with PBMCs and Ad-MSCs every 24 h (G3).

**Figure 4 vetsci-11-00636-f004:**
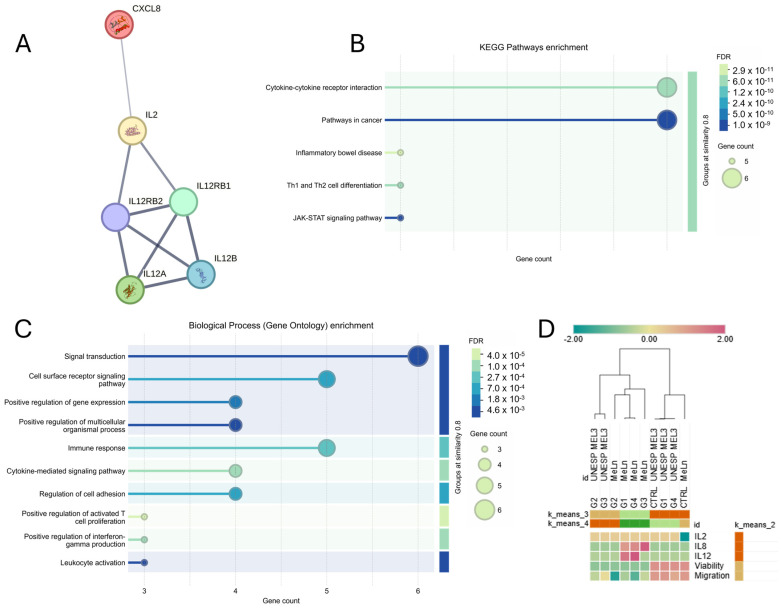
(**A**) Interactions observed among interleukins IL-2, IL-8, and IL-12 using the STRING tool. The edges represent interactions, with edge thickness indicating the confidence edge of each interaction. (**B**,**C**) Pathway enrichment analysis for the studied interleukins using STRING, showing a strong association with cytokine–cytokine receptor interactions, cancer-related pathways, and immune response activation signaling. FDR: False Discovery Rate. (**D**) Heatmap and clustering of the groups across different cell lines. In the k-means clustering, G2 groups were more closely correlated across cell lines, while G1, G3, and G4 (MeLn) groups were more closely related to each other. Additionally, the control, G1, and G4 (UNESP MEL3) groups clustered together. Hierarchically, MeLn groups were clustered closely together, as were UNESP MEL 3 groups, except for the MeLn control, which clustered closer to the CTRL, G1, and G4 groups of UNESP-MEL3 (Z-score scale).

**Figure 5 vetsci-11-00636-f005:**
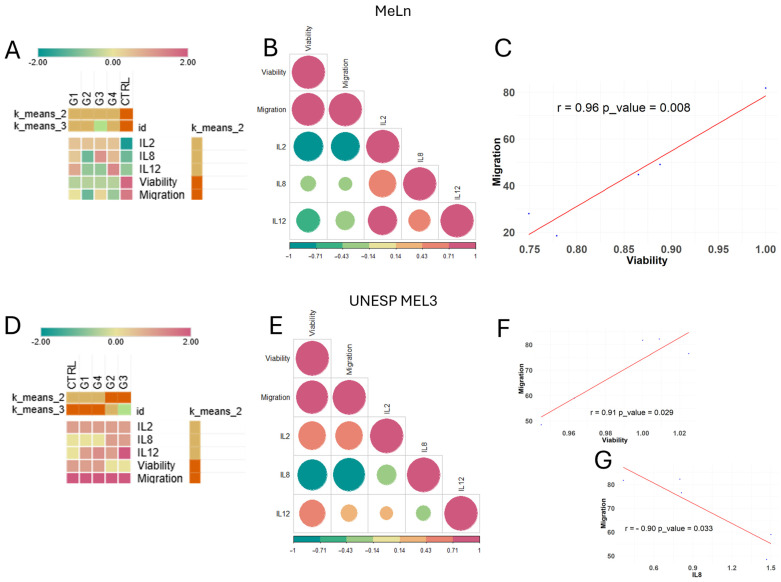
Heatmap, clustering, and correlation analysis of groups for each cell line. (**A**) MeLn cell line k-means clustering indicates that treated groups are more closely correlated than the control group (Z-score scale). (**B**) Correlation matrix. A graphical representation of correlation values among analyzed parameters, where the color scale indicates the R-value. Circle size represents correlation strength, with red tones indicating positive correlations and green tones indicating negative correlations. A significant correlation was observed only between cell viability and migration capacity. (**C**) Scatter plot. Displays a strong positive correlation between cellular migration and viability (r = 0.96; *p* = 0.008). (**D**) UNESP MEL3 cell line k-means clustering shows G2 and G3 as closely related, while CTRL, G1, and G4 display greater correlation among themselves. (**E**) Correlation matrix. Graphical representation of the correlation among analyzed parameters, with significant correlations observed between viability and migration capacity, as well as IL-8 expression and migration capacity. (**F**) Scatter plot. It shows a strong positive correlation between cellular migration and viability (r = 0.91; *p* = 0.029). (**G**) Scatter plot. Displays a strong negative correlation between cellular migration and IL-8 expression (r = −0.9; *p* = 0.033).

**Table 1 vetsci-11-00636-t001:** Sequences of the primers used for RT-qPCR of different interleukins.

Interleukin	Forward	Reverse
*IL-2*	5′-CAA CTC CTG CCA CAA TGT ACA AA-3′	5′-TGC GAC AAG TAC AAG CGT CAG T-3′
*IL-8*	5′-TGT TGC TCT CTT GGC AGC TTT-3′	5′-TTG ACA GAA CTG CAG CTT CAC A-3′
*IL-12*	5′-AGG CAG ATC TTT CTG GAT CAA AA-3′	5′-TCA GGG CCT GTA

## Data Availability

The raw data that support the findings of this study are available from the corresponding author upon reasonable request.

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
