# Peer review of "Canine Adipose-Derived Mesenchymal Stromal Cells Reduce Cell Viability and Migration of Metastatic Canine Oral Melanoma Cell Lines In Vitro"

_vetsci, 2024, doi:10.3390/vetsci11120636_

Round 1
Reviewer 1 Report
Comments and Suggestions for Authors
In this paper, the authors investigated the potential inhibitory effects of canine adipose-derived mesenchymal stromal cells (MSCs) on two melanoma cancer cell lines developed from metastatic canine oral melanoma (MeLn and UNESP-MEL3). The study examined also the immunomodulation of MSCs on peripheral blood mononuclear cells (PBMCs). The authors focused on the indirect effects of MSCs, alone or in association with PBMCs and evaluated the viability and migratory behavior of canine oral melanomas. They studied by gene expression, the levels of IL-2, IL-8 and IL-12 in MSCs and PBMCs. The authors observed that MSCs reduced cancer cell migration and survival but these effects varied depending on the cancer cell type. Furthermore, MSCs modified IL expression profiles in co-cultured cells. They conclude that the modulation of the tumor microenvironment may have therapeutic benefits for canine oral melanoma but further research is need.
Overall the paper is interesting and the results are clearly presented. The tables and figures are clear, very complete and the conclusions of this paper are, I think, justified. The paper appears of interest for the journal’s readership. The manuscript needs several revisions. The authors can consider following suggestions to improve their research paper.
As reported by the authors in the introduction, cancer has recently emerged as a promising target for cell-based therapies with MSCs playing a potential role in modulating the tumor microenvironment by expressing various cytokines and cellular interactions. However, MSCs not only have a strong tropism for wounds and damaged tissues but also for tumors. It seems MSCs play a dual role, as they have a double-edged sword effect on cancer. The pro- and anti-tumorigenic behavior depends on the balance between their pro- and anti-inflammatory phenotypes. It has been proposed that MSCs, like monocytes, can be polarized by TLR challenging in two homogenously acting phenotypes, MSC1 and MSC2. Different studies described the effects of MSC1 and MSC2 on tumor growth and spread. The co-culture of MSC1 with various cancer cells diminished their growth in contrast to growth promoting MSC2 co-cultures (Waterman et al, Plos One 2012; Rivera-Cruz et al, Cytotherapy 2023). The authors should add this important point in the introduction.
Different experiment groups were designed to evaluate whether MSCs and/or PBMC could modulate the effects on melanoma cells. Indeed MSCs have demonstrated significant immunomodulatory properties. MSCs have the ability to regulate the function of most of the effector cells in the immune response by direct contact or through soluble factors, extracellular vesicles or a synergy of these tools.
However, some authors claim that MSCs are not inherently immunosuppressive and need to be activated in order to display full capacities. Cell priming (hypoxia, inflammatory mediators, cytokines, pharmacological drugs, 3D culture, allows to obtain MSCs with enhanced immunosuppressive potential. Interestingly, some authors have tested to culture MSCs in direct or indirect contact with PBMC. In these conditions, MSCs significantly increased the expression of IL-8, IL-6, TNF-α. When PBMC were activated by a mixed lymphocyte reaction (MLR), changes were identified in the morphology, gene expression and function of MSCs. What would be the impact of PBMC activation on the inhibition of melanoma cells by MSCs. This aspect would be interesting to evaluate.
Author Response
Dear Editor and reviewers,
We are submitting a revised version of the manuscript Canine Adipose-Derived Mesenchymal Stromal Cells Reduce Cell Viability and Migration of Metastatic Canine Oral Melanoma Cell Line In Vitro. We appreciate the time and effort you and the reviewer have devoted to providing your valuable comments on the manuscript. Your suggestions have been of great value to us. We have fully considered and implemented whenever as possible all suggestions and modifications proposed.
We truly hope this revised version has properly addressed all concerns and may be acceptable for publication. Here is a point-by-point response to the reviewers’ comments and concerns:
Comment: As reported by the authors in the introduction, cancer has recently emerged as a promising target for cell-based therapies with MSCs playing a potential role in modulating the tumor microenvironment by expressing various cytokines and cellular interactions. However, MSCs not only have a strong tropism for wounds and damaged tissues but also for tumors. It seems MSCs play a dual role, as they have a double-edged sword effect on cancer. The pro- and anti-tumorigenic behavior depends on the balance between their pro- and anti-inflammatory phenotypes. It has been proposed that MSCs, like monocytes, can be polarized by TLR challenging in two homogenously acting phenotypes, MSC1 and MSC2. Different studies described the effects of MSC1 and MSC2 on tumor growth and spread. The co-culture of MSC1 with various cancer cells diminished their growth in contrast to growth promoting MSC2 co-cultures (Waterman et al, Plos One 2012; Rivera-Cruz et al, Cytotherapy 2023). The authors should add this important point in the introduction.
Answer: Thank you so much for your comment. We included a brief paragraph in the introduction highlighting the presence of these phenotypes in human MSCs (lines 80–85). Additionally, we discussed the need for evaluating this phenotypic profile in canine samples (lines 397–404).
Different experiment groups were designed to evaluate whether MSCs and/or PBMC could modulate the effects on melanoma cells. Indeed MSCs have demonstrated significant immunomodulatory properties. MSCs have the ability to regulate the function of most of the effector cells in the immune response by direct contact or through soluble factors, extracellular vesicles or a synergy of these tools.
However, some authors claim that MSCs are not inherently immunosuppressive and need to be activated in order to display full capacities. Cell priming (hypoxia, inflammatory mediators, cytokines, pharmacological drugs, 3D culture, allows to obtain MSCs with enhanced immunosuppressive potential. Interestingly, some authors have tested to culture MSCs in direct or indirect contact with PBMC. In these conditions, MSCs significantly increased the expression of IL-8, IL-6, TNF-α. When PBMC were activated by a mixed lymphocyte reaction (MLR), changes were identified in the morphology, gene expression and function of MSCs. What would be the impact of PBMC activation on the inhibition of melanoma cells by MSCs. This aspect would be interesting to evaluate.
Answer: Thank you so much for this suggestion and we will consider it in the future. Unfortunately, we currently lack the resources for an analysis of this scope. However, we added a paragraph discussing this aspect and suggesting future studies to explore this analysis (lines 457–462).

Reviewer 2 Report
Comments and Suggestions for Authors
In this manuscript, the authors tested whether combining mesenchymal stem cells (MSCs) with another type of immune cell could inhibit canine oral melanoma (COM) cell viability and migration. They also examined three key interleukins—2, 8, and 12—associated with immune responses to understand how they might reduce cancer cell growth and migration. Although the effects varied depending on the cancer cell type and only two lines were studied, their findings suggest that MSCs could have therapeutic potential for COM by inhibiting cell migration and reducing viability.
I have only a few minor points:
1. One of the limitations of the study is the use of MSCs from only one donor. If the authors could add data from more donors that would increase the value of the results.
2. Line 145 (Figure 1 legend). What authors refer as ‘a distance of 0.4 um’ is actually the size of the pore of the transwell membrane, which prevents . This is confusing and should be clarified.
3. Lines 204-206. How did the authors count the number of migrated cells? Did they count all the cells in the transwells or only in a number of fields? Authors should be more concise on this subject because the variability in the experiment is very low in some groups and very high in others (Figure 2 B and D).
4. Figure 2. Increase the font size because it is barely readable (check also Figure 3 and 5C, 5F and 5G). Statistical significance is usually indicated in the figure by using one or several asterisks depending of the p-values, not in the legend (check also Figure 3). It is hard to believe that in the ‘D’ panel, values from group 2 (MSCs) are not statistically significant compared to the control, especially because in the ‘B’ panel this group shows the highest differences. The authors may review the data or check if they are using the right statistical test (compare with others).
5. Figure 4. The authors state that only the groups treated with PBMCs showed increased IL-2 expression. As they have isolated PBMCs from several donors, is this true for all the donors tested?
Author Response
Dear Editor and reviewers,
We are submitting a revised version of the manuscript Canine Adipose-Derived Mesenchymal Stromal Cells Reduce Cell Viability and Migration of Metastatic Canine Oral Melanoma Cell Line In Vitro. We appreciate the time and effort you and the reviewer have devoted to providing your valuable comments on the manuscript. Your suggestions have been of great value to us. We have fully considered and implemented whenever as possible all suggestions and modifications proposed.
We truly hope this revised version has properly addressed all concerns and may be acceptable for publication. Here is a point-by-point response to the reviewers’ comments and concerns:
- One of the limitations of the study is the use of MSCs from only one donor. If the authors could add data from more donors that would increase the value of the results.
Answer: Unfortunately, we do not have the resources to conduct further evaluations using MSCs derived from other animals. However, this is indeed an important aspect to be considered. We have added a sentence highlighting the importance of evaluating differences in MSCs with distinct phenotypes, as they may produce opposing results (lines 397–404).
- Line 145 (Figure 1 legend). What authors refer as ‘a distance of 0.4 um’ is actually the size of the pore of the transwell membrane, which prevents. This is confusing and should be clarified.
Answer: we apologize for the confusion in this sentence. Indeed, the sentence was unclear. We have restructured it to avoid misinterpretations (lines 152–153).
- Lines 204-206. How did the authors count the number of migrated cells? Did they count all the cells in the transwells or only in a number of fields? Authors should be more concise on this subject because the variability in the experiment is very low in some groups and very high in others (Figure 2 B and D).
Answer: We have added more details regarding the methodology used for cell counting in this assay. Specifically, for each experimental replicate, cell counts were performed in four areas of the transwell membrane, and the total count for each replicate was used in the statistical analysis.
- Figure 2. Increase the font size because it is barely readable (check also Figure 3 and 5C, 5F and 5G). Statistical significance is usually indicated in the figure by using one or several asterisks depending of the p-values, not in the legend (check also Figure 3). It is hard to believe that in the ‘D’ panel, values from group 2 (MSCs) are not statistically significant compared to the control, especially because in the ‘B’ panel this group shows the highest differences. The authors may review the data or check if they are using the right statistical test (compare with others).
Answer: Thank you for your comment. We agree that some images were indeed difficult to interpret. To address this, we increased the font size across all figures to improve visualization.
Regarding the use of asterisks, while we initially considered them for the images, the inclusion of multiple comparisons—not only between groups and the control but also among the groups themselves—made the use of asterisks overly complex and visually confusing. Therefore, we opted for letters which provided better clarity and understanding of the figures.
Lastly, although it is visually apparent that Group 2 shows a statistical difference, the p-value, despite being close to 0.05, remained above the threshold (p=0.06) according to Tukey’s multiple comparison test. This result is discussed (lines 413-420), and we consider it indicative of a potential trend in MSCs' ability to reduce migration in both tumor types.5.
Figure 4. The authors state that only the groups treated with PBMCs showed increased IL-2 expression. As they have isolated PBMCs from several donors, is this true for all the donors tested?
Answer: The PBMCs from individual donors were not analyzed separately, instead, a pooled sample of these cells was used for analysis. Therefore, we cannot provide specific insights regarding individual donor variability. However, we observed a statistically significant increase in IL-2 levels compared to the control, particularly in groups treated with PBMCs at any stage of the experiment.
Figure 4 (A, B, and C) illustrates previously established interactions between these interleukins and their roles in various signaling pathways. In contrast, Figure 4D shows the correlations among the different cell groups. Notably, the two studied cell types displayed distinct and often divergent behaviors across most groups, indicating that each cell type responded uniquely to the experimental conditions.
